



# Annual Cycle of Hygroscopic Properties and Mixing State of the Suburban Aerosol in Athens, Greece.

Christina Spitieri[1], Maria Gini[1], Martin Gysel-Beer[2] and Kostas Eleftheriadis[1]

[1] Environmental Radioactivity Laboratory, Institute of Nuclear and Radiological Science & Technology, Energy & Safety,
NCSR Demokritos, 15310 Ag. Paraskevi, Athens, Greece
[2] Laboratory of Atmospheric Chemistry, Paul Scherrer Institute, Forschungsstrasse 111, Villigen PSI, Switzerland

*Correspondence to*: Christina Spitieri (spitieri@ipta.demokritos.gr) and Maria Gini (gini@ipta.demokritos.gr)

**Abstract.** The hygroscopic properties of atmospheric aerosol were investigated at a suburban environment in Athens, Greece,
from August 2016 to July 2017. The Growth Factor Distribution Probability Density Function, (GF-PDF), and mixing state
were determined with a Hygroscopicity Tandem Differential Mobility Analyzer, (HTDMA). Four dry particle sizes, ($D_0$), were
selected to be analyzed in terms of their hygroscopic properties at 90 % relative humidity. The annual mean GFs for $D_0$ =30,
50, 80, and 250 nm, were found to be equal to 1.28, 1.11, 1.14, and 1.22 respectively. The hygroscopic growth spectra can be
divided into two distinct hygroscopic ranges; a non or slightly hygroscopic mode (GF<1.12) and a moderately hygroscopic
mode (GF>1.12), which are representative of a suburban environment influenced by local/regional emissions and background
aerosol. The standard deviation σ of the GF-PDF was employed as a measure of the mixing state of ambient aerosol. The 30
nm particles were mostly internally mixed, whereas larger particles were found to be externally mixed, with either a distinct
bimodal structure or with partly overlapping modes. Cluster analysis on the hourly dry number size distributions measured in
parallel, provided the link between aerosol hygroscopicity and growth/evaporation dynamics. The size distributions were
classified into five groups, with the "mixed, urban and aerosol background" (67%) and "urban-nocturnal" aerosol (12%) to
account for 79% of the results. The hygroscopic properties for 50 nm and 80 nm were found to be similar in all cases, indicating
particles of similar nature and origin across these sizes. This was also confirmed through the modal analysis of the average
number size distributions for each cluster; the 50 nm and 80 nm particles were found to belong to the same Aitken mode in
most cases. The 250 nm particles (i.e. accumulation mode) were generally more hygroscopic than Aitken particles, but less
hygroscopic than the 30 nm particles (nuclei mode).

## Introduction

Atmospheric aerosol particles in the ambient atmosphere affect the radiation budget of the planet and the regional and global
climate (IPCC, 2013; Rosenfeld et al., 2014), through direct and indirect effects (Li et al., 2016). As a direct effect, aerosol
particles interact with solar radiation through light absorption and scattering, inducing a positive or negative radiation forcing,



respectively (Haywood and Boucher, 2000). As an indirect effect, aerosol particles can act as cloud condensation nuclei (CCN) affecting cloud's microstructure and lifetime. The climate-relevant properties of atmospheric aerosol particles are largely determined by their size, chemical composition, hygroscopicity and state of mixing (Zhang et al., 2011; Kaufman et al., 2002; McFiggans et al., 2006).

The hygroscopic properties of atmospheric particles are strongly related to particle chemical composition (Gunthe et al., 2009; Gysel et al., 2007), while they undergo continuous changes over particle lifetime. Research results have shown that the relative non-hygroscopic fresh organic aerosol can be converted to become hygroscopic through physical and chemical atmospheric processes (Kanakidou et al., 2005). This is also the case for soot particles originating from different sources (e.g. biomass burning, diesel soot). Condensation of secondary species on the particle phase and on-going coagulation,  may alter the

hygroscopic properties of soot particles from non-hygroscopic, to hygroscopic with the latter capable to act as CCN (Tritscher et al., 2011; Kotchenruther and Hobbs, 1998). Motos et al., (2019) also quantified the link between the black carbon core size and mixing state and droplet activation.

Aerosol hygroscopic growth can be measured or estimated by both direct and indirect techniques (Dean A. Hegg et al., 2007, P. Achtert et al., 2009). The most widespread real-time direct measurement technique for fine mode aerosol is the

Hygroscopicity Tandem Differential Mobility Analyzer, (HTDMA), which determines the growth factor, (GF), of particles at a given dry particle diameter and relative humidity, (RH). Then, the hygroscopic parameter κ can be calculated as described by Peter and Kreidenweis (2007), which allows to infer basic information on the chemical composition of the aerosol, though typically not fully unambiguous (Bezantakos et al., 2013; Gysel et al., 2007). For typical atmospheric soluble salt particles such as ammonium sulfate or sodium chloride, the value of κ is 0.53 and 1.12, respectively, whereas for secondary organic

aerosols (SOAs) the κ value typically ranges between 0.0 and 0.2 (Rickards et al., 2013; Rose et al., 2009).

The cloud nucleating effectivity of aerosol particles and the overall aerosol-cloud-climate interactions, depend on the distribution of components among individual particles, termed aerosol mixing state (Riemer et al., 2019). Ambient aerosol is usually considered a heterogeneous mixture of particles with different chemical compositions and sizes. We refer to an internal mixture of aerosol when the particles of the same size have similar chemical composition. Whereas, in an external

mixture particles of the same size have distinctly different chemical composition. The aerosol in urban/sub-urban environments typically is an external mixture of non-hygroscopic aerosol from fresh local emissions and moderately hygroscopic background aerosol (Wang et al., 2018; Enroth et al., 2018; Swietlicki et al., 2008 ), whereas in marine environments the aerosols tend to be internally mixed (Massling et al., 2007).

Long-term measurements of aerosol hygroscopicity are useful to better identify the link between particle hygroscopic growth

with respect to particle emission sources, formation and transformation processes, with few detailed studies available so far (Kammermann et al., 2010, Fors et al., 2011, Sellegri et al., 2014). The present study aims at providing insights into the hygroscopic properties and state of mixing of ambient aerosol and the origin of ambient ultrafine and fine particles, through 1 year of measurements of key microphysical parameters (i.e. size distributions and time- and size- resolved HTDMA data) in suburban environment. The hygroscopicity of ambient aerosol was investigated in the particle size range between 30 and 250



nm, providing information about the month-to-month variability, seasonal cycle and diurnal pattern of hygroscopicity for selected particle sizes of the suburban aerosol in Athens.

## 2. Methodology

### 2.1 Sampling Site


The measurement campaign was conducted from August 2016 to July 2017, at the Demokritos station (fig.1.), member of GAW and part of the ACTRIS and PANACEA infrastructures within the National Centre of Scientific Research Demokritos campus, a vegetated area at the foot of Mount Hymettus, about 8 km to the North east from Athens city centre. It is an urban background station, representative of the atmospheric aerosol in the suburbs of the Athens Metropolitan Area. The site is

partially influenced by transported pollution from the urban area of Athens (Eleftheriadis et al., 2021) (i.e. under most atmospheric conditions) and partially by the incoming regional aerosol (i.e. under Northern or Eastern winds).

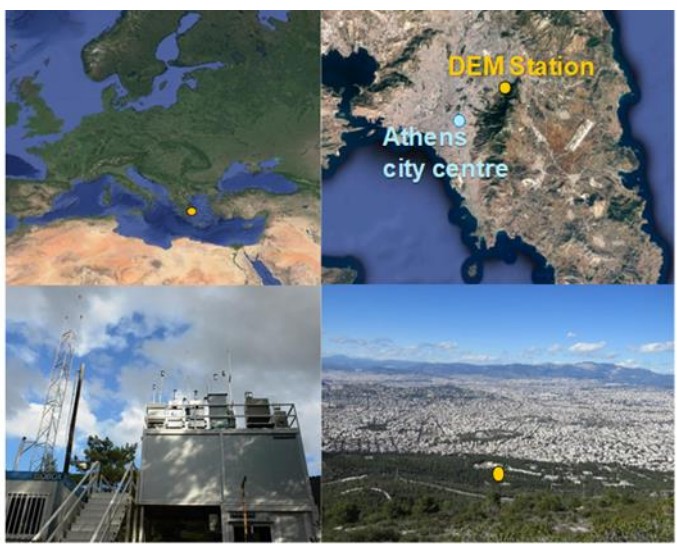

**Figure 1** The Demokritos Atmospheric Aerosol Measurement station in Athens, (from © Google Maps).

### 2.2 Instrumentation

A custom-built Humidified Tandem Differential Mobility Analyzer, (HTDMA), was used to measure the hygroscopic growth factor distributions of ambient aerosol particles with selected dry diameters $D_0$ at certain narrow size fractions centered around 30 nm, 50 nm, 80 nm and 250 nm, exposed at relative humidity (RH) of $90 \pm 2$ %. Figure 2 shows the main components of the

HTDMA system. The HTDMA consists of two differential mobility analyzers (DMAs) in tandem mode, a humidifier section and a condensation particle counter (CPC 3772, TSI). The polydisperse aerosol was initially dried, passing through an aerosol Nafion dryer, and brought to charge equilibrium passing through a $^{85}$Kr bipolar neutralizer, before entering the first DMA,





(DMA₁), where the specific particle sizes were selected (monodisperse aerosol) according to their electrical mobility. Then, the monodispersed and dried particles were conditioned, by passing through the humidifier section, at a well-defined relative

humidity (set point 90 %), before entering the second DMA (DMA₂); the sheath flow of the DMA₂ was also humidified at a relative humidity of 90 %. The DMA₂ was operated in line with the CPC in a scanning particle mobility sizer configuration, (SMPS), to measure the particle size distribution of the conditioned wet aerosol.

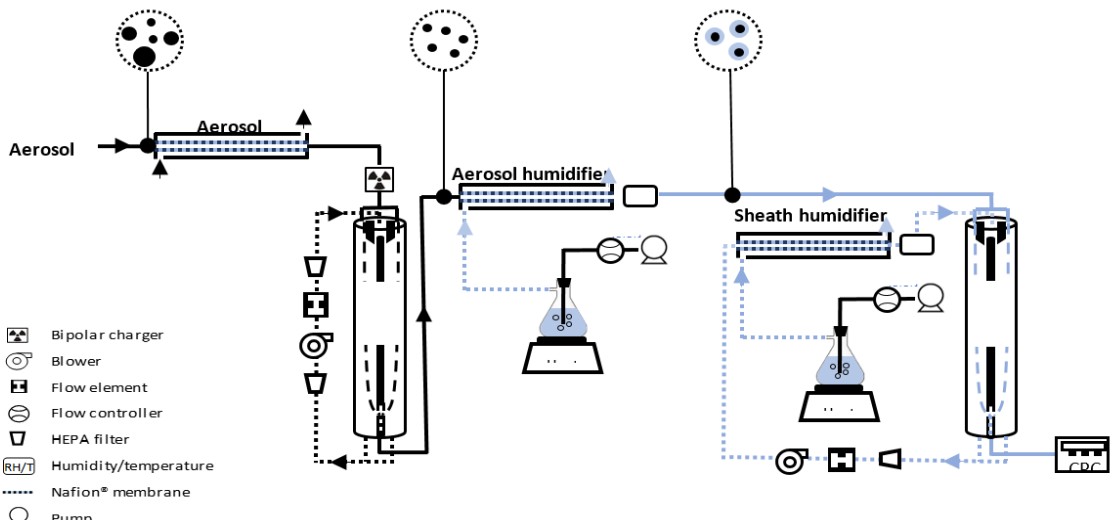

**Figure 2** Schematic diagram of the HTDMA system


The aerosol number size distributions of ambient aerosol (dry) were measured in parallel by the standard SMPS system of the Athens ACTRIS station, with five-minute time resolution, consisting of an electrostatic classifier (TSI Inc. model 3080), a cylindrical differential mobility analyzer column, (TSI Inc, model 3081), and a condensation particle counter, (TSI Inc., model 3772). The SMPS was operated at an aerosol flow rate of 1 lpm and a sheath flow rate 5 lpm, extending the measured particle

size range from 10 nm to 550 nm. Both, aerosol and sheath, SMPS flows were dried to relative humidity lower than 45 % using a Nafion drier. Data acquisition and analysis was performed using the non-commercial TROPOS-SMPS data evaluation software (Wiedensohler et al., 2012). To achieve the highest measurement accuracy with SMPS measurements, the technical recommendations and quality control procedures proposed by Wiedensohler et al. (2012) were followed.

The standard meteorological parameters (i.e. temperature, humidity, wind speed, wind direction and solar radiation) were

recorded at an hourly time interval. The meteorological sensors were installed on a meteorological mast, at 10 m height above ground.



## 2.3 Data analysis

### 2.3.1 HTDMA data inversion and fitting procedure

Due to water uptake, the diameter of the humidified particles ($D$(RH)) increase, and the ratio between humidified, ($D$(RH)), and dry particle diameter, $D_0$, is defined as the Growth Factor (GF):

$$GF = \frac{D(RH)}{D_0},\qquad\qquad(1)$$

where $D$(RH) is the particle diameter at the given RH and $D_0$ is the dry diameter selected by the first DMA. The particle concentration at the HTDMA outlet as a function of growth factor (GF) set at the HTDMA is referred to as Measurement Distribution Function, (MDF).

Then, an inversion algorithm applied to the measured MDF to retrieve the actual growth factor probability density functions (GF-PDF), which describe the probability that a particle with a defined dry size exhibits a certain GF at the specified relative humidity. A key element for the data inversion is the Kernel function, which describes the physics of the HTDMA instrument. The Kernel width calibration, for particles exhibiting a true growth factor of 1.0, of the HTDMA and data inversion, using TDMAinv algorithm, were performed according to the methodology described by Gysel et al. (2009). The underlying principle of TDMA inversion approaches is to find an inverted GF-PDF, such that a minimum $x^2$-residual is obtained between the measured MDF and the reconstituted MDF (R-MDF). The growth factor probability density functions (GF-PDFs) were afterwards normalized to unity. The GF-PDFs measured in the range 87 % < RH < 92 % were recalculated to RH = 90 % following the procedure described by Gysel et al. (2009), in order to reduce uncertainties associated with RH variations. Each GF-PDF was described as a piecewise linear function with the midpoint of the first and last inversion bin at GF=0.7 and GF=2.5, respectively, and a resolution of ΔGF=0.1. The GF standard deviation, σ, of a GF-PDF, was determined according to Eq. (C.6) in Gysel et al. (2009).

The σ is used as a measure for the spread of growth factor to describe the mixing state (Sjogren et al., 2008).

In the present study, the inverted data can be grouped into three cases, representative of the aerosol mixing state. Indicative inverted GF-PDFs are presented in Fig. 3., reflecting the three cases of mixing state. Specifically, σ ≤0.07 indicates an internally mixed aerosol (Panel (A), σ ≥0.15 describes an externally mixed aerosol with two distinct modes (Panel (C), whereas GF-PDFs with 0.07 < σ <0.15 are considered as a continuum of mixing states with two overlapping modes or a broad mode Panel (B), (almost bimodal and externally mixed). The mean GF and the number fraction of particles of each mode was calculated by Eq. (C.9) and Eq. (C.8), respectively, in Gysel et al., 2009.

Additionally, the hygroscopicity parameter κ, was calculated as follows (Petters and Kreidenweis, 2007):

$$\kappa = \frac{(GF_3-1)(1-a_w)}{a_w}\qquad\qquad(2)$$





where $a_\mathrm{w}$ is the water activity, at which the growth factor was measured. According to Köhler theory (Köhler , 1936), $a_\mathrm{w}$ is obtained by


$$a_\mathrm{w} = \frac{RH}{exp(\frac{4\sigma_s v_w}{RTD})} \qquad (3)$$

where $\sigma_\mathrm{s}$ is the surface tension of the solution droplet (assumed to be pure water), $v_\mathrm{w}$ is the partial molar volume of water in solution, $R$ is the universal gas constant, $T$ is the temperature, and $D$ is the diameter of the droplet.

**Figure 3** Example of growth factor distributions measured at RH=90% of $D_0$=30nm particles (Panel (A), internally mixed), of $D_0$=80nm particles (Panel (B), continuum of mixing states) and of $D_0$=250nm particles (Panel (C), externally mixed with distinct modes). The red line






represents the measured particle counts, the blue line represents the reconstructed measured distribution function and the green line is the GF-PDF.


### 2.3.2 Cluster Analysis

k-means cluster analysis was applied to the hourly-average particle number size distributions to classify the distributions of the highest degree of similarity into the same cluster, reducing in that way the complexity of the dataset. The k-means method
aims to minimize the sum of the squared Euclidian distance between each dataset point and the corresponding cluster center (i.e. the mean of all the points in a cluster). Cluster analysis was performed using the IBM SPSS software. The interpretation of the origin of each cluster was based on the dominant size modes (Hussein et al., 2014), their hourly frequency of occurrence and the average values obtained for standard meteorological parameters.

### 2.3.3 Multimodal Analysis

The modal characteristics of the clustered average number size distributions were obtained by applying a curve-fitting algorithm as proposed by Hussein et al. (2005). The least squares method was used to best-fit the sum of up to 4 modes to the multi-modal distributions to the clustered distributions. The log-normal distributions were described by characteristic modal
parameters i.e. geometrical mean mobility diameter, number concentration and geometric standard deviation. Starting from an initial assumption, the modal parameters of each log-normal distribution were successively re-defined to obtain the best-fit curve. The algorithm starts by fitting a uni-modal log-normal distribution, and successively tests the possibility of increasing it to a bi-, a tri- and finally a tetra-modal distribution. The optimum best-fit curve was determined by minimizing the root mean square error (RMSE, %) as described by Vratolis et al. (2019).


## 3. Results and Discussion

### 3.1 Seasonal and monthly variability of aerosol GF and mixing state

The seasonal, as well as the annual, mean GF-PDFs, fig. 4., were calculated by averaging the individual GF-PDF for each dry
particle size, split by season. The GF-PDFs represent the mean distributions of growth factors of particles with $D_0$=30, 50, 80 and 250nm, which are representative of the nuclei mode, Aitken mode (50 and 80nm) and accumulation mode (250nm) respectively. However, some broadening of the modes in the GF-PDF and the shape of GF-PDF results from the calculation of averages. This does not necessarily provide a clear picture of the mixing state of these size fractions, when modal GF varies over time. The majority of the mean GF-PDFs of the nuclei mode particles with dry diameters of 30 nm were characterized by
a unimodal peak, (except from winter), with GFs ranged between 1.20 and 1.41. Non or slightly hygroscopic 30-nm-particles with GF~1.0 are essentially missing, in contrast to particles with $D_0$>30nm indicating that freshly emitted particles, such as





bare black carbon, are largely absent during most of the year with the exception of winter (January & February). Aging processes are very likely more efficient for the nuclei mode rather than the higher Aitken modes in modifying their hygroscopicity due to condensation of organics and inorganics onto the pre-existing particles (Vu et al., 2011).

**Figure 4** Seasonal and annual mean GF-PDFs for different dry particle sizes (30, 50, 80 and 250 nm)

The existence of a broad mode in the GF-PDF of 30nm particles in winter provides the expected aerosol phenomenology that both fresh (non or slightly hygroscopic), aged (moderately hygroscopic) emissions from traffic and other combustion sources





(biomass burning, residential heating) and hygroscopic fraction from the occasional nucleation events contribute to the 30 nm

fraction. The fact that all the above are visible only in the two winter months is the combined effect of the time scale of the

aging process and the time scale of mixing of fresh and background aerosol at the background location of the DEM station.

One must consider that during winter weaker aging results from the lowest temperatures and lowest photochemical activity

experienced in Athens. More detailed discussion for this fraction will follow further below.

The GF-PDFs of medium to large Aitken mode particles, i.e. at 50 nm and 80 nm, display a low degree of external mixing,

with average GF ranges between 1.00 and 1.23. The non or slightly hygroscopic and moderately hygroscopic particles make

almost equal contributions to the GF-PDF, indicating that the contribution of non or slightly -hygroscopic particles is higher

in the Aitken mode than the 30nm particles.

The separation of the two distinct hygroscopic modes (a bimodal distribution or a continuum of mixing states) is most

pronounced for the accumulation mode particles (250 nm). The contribution of the moderately hygroscopic mode to the total

hygroscopicity is higher compared to the hygroscopicity of Aitken particles. Particles larger than 100 nm are usually more

aged than the smaller particles, with higher values of GF (Cubison et al., 2006) and more immediately associated with the

atmospheric processing they undergo during long-range transport (Kalivitis et al., 2015). However, in winter time the expected

slowing down of secondary aerosol formation processes and the existence of larger primary particles, partly from biomass

burning (Bernardoni et al., 2017), make evident the distinct appearance of the fresh non or slightly hygroscopic mode and

moderately hygroscopic modes. Laborde et al., (2013) also showed that the particles with GF<1.12 themselves can be an

external mixture of BC rich and organic rich particles, in this peculiar case associated with local/regional traffic and wood

burning emissions probed at a peri-urban site of Paris. The background aerosol was found to be more hygroscopic in their case

compared to this study. In August, GF-PDFs were unimodal across the size range 50 nm $\leq D_0 \leq$ 250 nm, and the mean GF was

as low as 1.03 to 1.09. Such dominance of non or slightly hygroscopic particles suggest that the Aitken and accumulation mode

aerosol was dominated by non or slightly -hygroscopic BC potentially affected by nearby forest fires (Carrico et al., 2005).

The annual mean growth factors at 90 % relative humidity were found to be 1.28, 1.11, 1.14 and 1.22 for $D_0 =$ 30, 50, 80 and

250 nm, respectively. (fig. 5. Panel A). Although a distinct month to month variability was observed on the average GFs, there

was no distinct seasonal variability on the average GFs. It has to be empathized that  apparent features with higher values in

spring may partly be over-emphasized by the August feature, which likely is driven by a meteorological outlier rather than

being a representative feature of the aerosol typically encountered in August. In a further analysis, the uncertainty and error

analysis of the variation of GF percentiles of the GF-PDF, are presented as box plots in fig. 6, for each month, for the dry

diameters 30, 50, 80 and 250nm. Monthly mean values of hygroscopicity parameter κ, determined according to Equations (1)

and (2), were found to be 0.13, 0.05, 0.05 and 0.09 for particles with dry diameters 30, 50, 80 and 250 nm, respectively, with

a distinct month to month variability, (fig. 5. Panel B). Panel C in fig. 5. shows the monthly variation of the standard deviation,

σ, for all particle sizes, describing the mixing state of aerosol (Rose et al., 2010). The 30 nm particles are internally mixed for

all months with σ ≤ 0.07, except from January and February, where the mean σ was 0.10 and 0.13, respectively. In the latter

case the aerosol particles are externally mixed (overlapping modes). The particles in the Aitken mode (50 and 80 nm) appeared





to be characterized by a continuum of mixing states (externally mixed) with σ values ranging between 0.09 and 0.13, except from August, where the mean σ was 0.06 indicating an internally mixed aerosol. The $D_0$=250 nm particles were characterized by a high degree of external mixing ($0.09 \le \sigma \le 0.16$), during all months, except of August where the aerosol display a low degree of mixing state (σ=0.09). In August, the GF-PDFs were almost unimodal across the whole size range between 50 nm and 250 nm, and the mean GF was as low as 1.03 to 1.09. Such dominance of non or slightly hygroscopic particles suggest that the Aitken and accumulation mode aerosol might be dominated by mostly primary particles related to fresh carbonaceous

emissions.

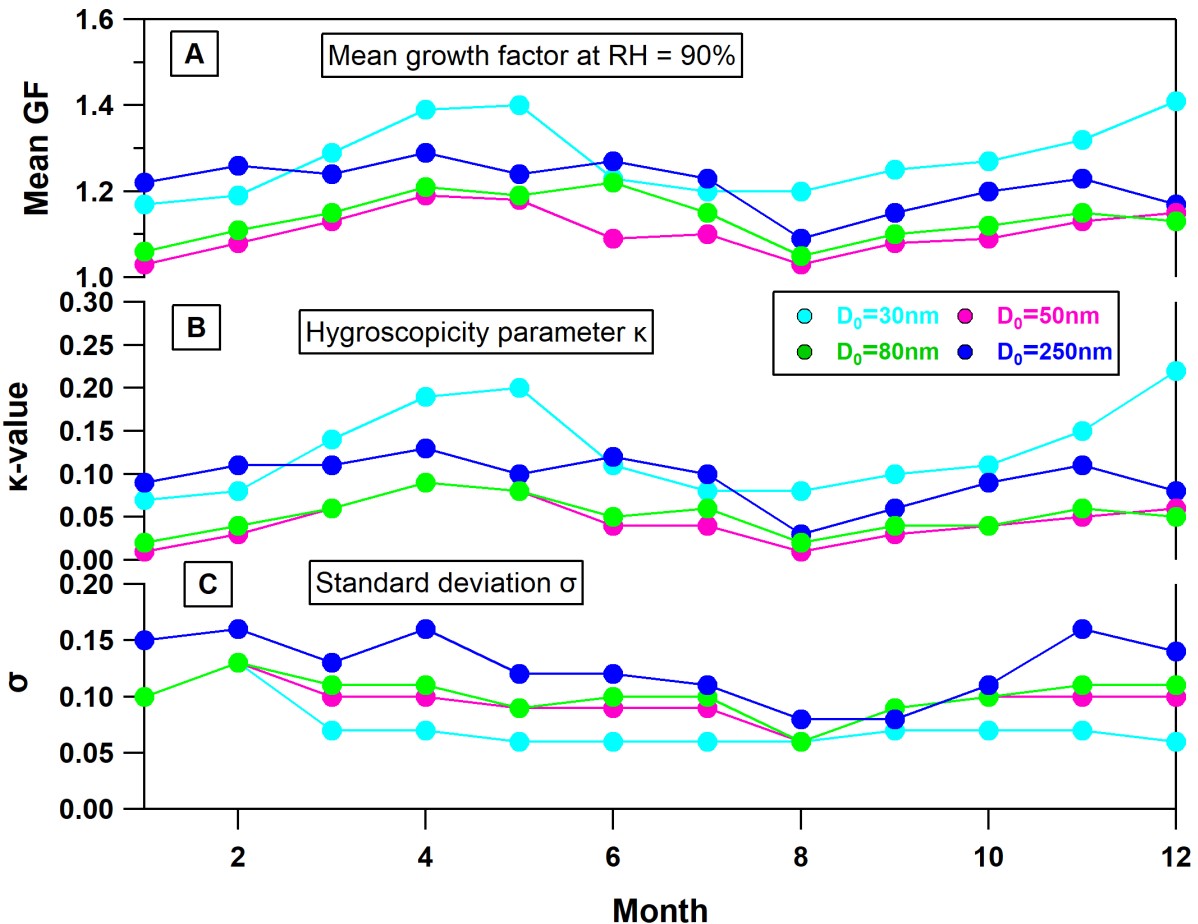

**Figure 5** Annual cycles A) of the mean growth factor (GF) at 90% RH, B) of the corresponding hygroscopicity parameter κ, C) of the monthly mean standard deviation σ, for different particle sizes

In Figure 6, the monthly boxplots of σ values are presented. For particle dry sizes $D_0 > 30$ nm, a distinct variability in the degree of mixing state was observed, from month to month. The highest spread of GFs was observed in February, implying the co-existence of traffic- and biomass burning- related emission sources. In contast, smaller σ values were observed in





August, indicating the existence of internally mixed aerosol in nuclei and Aitken modes, whereas the particles in the accumulation mode  remain externally mixed although with a lower degree of mixing.

In general, the Aitken particles and the particles in the accumulation mode ($D_0 > 30$ nm) and the particles in the accumulation mode can be characterized as an external mixture of moderately hygroscopic (aged) and non-hygroscopic aerosol (i.e. background aerosol mixed with regional and local emissions, respectively).

The bimodal GF-PDFs spectra, either with overlapping or with well-defined modes, was divided into two distinct ranges of particle hygroscopicity: one subset comprising non-hygroscopic and slightly hygroscopic particles with GF<1.12, the other

subset comprising moderately hygroscopic particles with GF>1.12. Mean properties of each subset were calculated by averaging GF-PDFs above/below this threshold GF of 1.12. This threshold GF coincides with the typical local minimum in the GF-PDFs observed in this study, which also is in line with the findings of previous studies (Kim et al., 2020).  Note, sensitivity analyses performed by changing the selected threshold GF of 1.12 to 1.20 had a very little effect on the calculated parameters. The particles with GF below/above the threshold can be present as distinct GF-modes, often the case for the

accumulation mode size range, or partially to fully overlapping GF-modes, often the case for the smaller sizes within the Aitken mode.







**Figure 6** Percentiles of the GF-PDF and standard deviation, σ, for each dry size and for each month. Box plots with whiskers show $10^{th}$, $25^{th}$, $50^{th}$, $75^{th}$ and $90^{th}$ percentiles.


**Figure 7** Annual cycles A) of the monthly mean GFs for non- or slightly hygroscopic particles at 90% RH, B) of number fraction of non- and/or slightly hygroscopic particles at 90% RH, C) of the monthly mean GFs for moderately hygroscopic particles at 90% RH, and D) of number fraction of moderately hygroscopic particles, for different particle sizes.




Figure 7 shows the monthly mean GFs and the number fraction, $f_{GF}$, of non or slightly hygroscopic and moderately hygroscopic mode, for each $D_0$. The variation of the mean GF for particles with GF < 1.12 is expected to be relative low, given that the upper boundary of the non or slightly hygroscopic mode (1.12) is quite low and close to the lowest boundary (1.0). Whereas, the mean GF of the moderately hygroscopic mode presented larger variability, with the minimum to be observed in summer

(i.e. August) for particles with $D_0 > 30$ nm. In short, the mean GF of the subset with GF>1.12 was larger for the 30 nm and 250 nm particles compared with the 50 and 80 nm particles. The different hygroscopic properties of these particles reflect their different chemical compositions, with the nuclei particles and the particles in the accumulation mode containing a larger fraction of more hygroscopic compounds, such as inorganics and more oxidized organics (Bougatioti et al., 2016), than that of Aitken particles. This trend in the size dependence of the average GFs for the Aitken and accumulation modes is in line with

the results from previous studies (Xu et al., 2019; Juranyi et al., 2013; Petäjä et al., 2007). Indicative GFs values are listed in Table 2 for different chemical compound classes. Table S1 summarizes the monthly mean GFs at RH=90 %, the mean GF of each mode, along with the hygroscopicity parameter κ and the number fraction of each mode, for dry diameters 30, 50, 80 and 250 nm.

The number fraction of each mode also significantly varied from month to month for all dry sizes, with distinct variability in

the relative contributions of particles with small or moderate-to-large growth factors. Specifically, for dry particle diameters $D_0 > 30$ nm, the contribution of the non- and/or slightly hygroscopic mode was maximum in spring and minimum in winter. In the case of Aitken particles, the non-hygroscopic particles almost equal contributed to aerosol hygroscopicity with the slightly hygroscopic particles in all seasons except for spring. This implies higher contribution of fresh and regional aerosol in the Aitken mode compared to the accumulation mode. For particles with $D_0 = 250$ nm, the moderately hygroscopic particles

clearly dominate over those with GF<1.12 for all seasons. Specifically, the average number fraction of the slightly hygroscopic particles were 0.62, 0.80, 0.67 and 0.70 in winter, spring, summer and autumn, respectively. This is consistent with the perception that the accumulation mode is dominated by aged aerosol (i.e. background aerosol), which is typically considered more hygroscopic than fresh emitted particles (Psichoudaki et al., 2018).

**TABLE 2** Mean Growth Factors measured at RH=90% for cold and warm period

| Chemical Composition | Growth Factor, (GF) | Source |
|---|---|---|
| **BC, Mineral Dust** | **<1.05** | **Vlasenko et al., 2005** |
| **Biomass Burning** | **1.15-1.65** | **Cocker et al., 2001** |
| **Aged wood smoke** | **1.3-1.5** | **Kotchenruther and Hobbs, 1998** |
| **Fresh wood smoke** | **1.1-1.3** | **Kotchenruther and Hobbs, 1998** |
| **Inorganic Ions** | **~1.7** | **Gysel et al., 2002** |





| Organic Compounds | 1.0-1.7 | Koehler et al., 2006 |
|---|---|---|

## 3.2 Annual and seasonal diurnal variability

The mean diurnal variability of critical meteorological parameters i.e. wind speed, temperature and relative humidity are shown
in fig. S2. The daily average temperature varied between 17.2 °C and 20.5 °C, peaking at midday. The daily average wind
speed varied between 1.81 ms⁻¹ and 2.84 ms⁻¹ peaking also at midday. The atmosphere was relatively wet with an average daily
relative humidity ~ 60% and a peak at night and early morning (~ 68%) and a minimum at midday (~ 49%). Generally, higher
concentrations of traffic-related pollutants are expected to be observed at the suburban site at midday, when conditions favor
mixing and dispersion of the generated aerosol across the Athens valley. The prevailing  westerly winds are stronger resulting
in a well-mixed atmosphere, while the pollutants are transported from the city to the suburban site (Kalogridis et al., 2018).
During the evening hours, concentrations peak due to the developing local inversion/nocturnal boundary layer and to additional
emissions by residential heating in winter (Eleftheriadis et al., 2021).

In general, particle hygroscopicity was lower during morning rush hours when a peak in the traffic-related emissions is
expected to occur, while during the afternoon an increase in hygroscopicity was observed which may be attributed to the
condensation of water soluble organics and inorganics on fresh primary particles of local/regional origin and on  background
aerosol (Psichoudaki et al., 2018). According to previous studies, the mass fraction of secondary organic compounds, which
is generally higher in summer, peaks at noon (Diapouli et al., 2017), resulting in intermediate GF values (GF < 1.3) (Bourcier
et al., 2012).

Mean GF, standard deviation, σ, GFs and number fractions of the non-hygroscopic and the slightly hygroscopic modes are
presented in fig. 8., for all particle's sizes. The Aitken particles are generally externally mixed throughout the day, with higher
contribution of the moderately hygroscopic mode in the early morning (03:00-05:00). A second but less pronounced peak also
appeared in the afternoon. The 24h cycle of the mean GF of the non or slightly hygroscopic mode (GF < 1.12) appeared to
have two peaks, with the major contribution of the non-hygroscopic particles to occur at midday. Particles in the accumulation
mode appeared to have a somewhat similar hygroscopic behavior, in terms of diurnal variability, with Aitken particles, even
though with less pronounced changes within the day for both growth factors and number fractions.









**Figure 8** Diurnal variation of A) mean GF, measured at RH=90% , B) standard deviation of the mean GF, C) GF of non or slightly hygroscopic particles, D) the number fraction of non or slightly hygroscopic particles, E) GF of moderately hygroscopic particles, and F) the number fraction of moderately hygroscopic particles

For the 30 nm particles, it was observed that the GF of the slightly hygroscopic mode was higher (GF > 1.3) between late evening and early morning (00:00 – 05:00 UTC+2), when the relative humidity appeared to have the maximum values, as well as at early afternoon (15:00 – 20:00 UTC+2), whereas the minimum appeared at noon (GF < 1.3) (21:00). At the DEM station, the 30 nm particles are generally related to traffic emissions or new particle formation (Vratolis, et al., 2019). As the particles undergo atmospheric aging their composition changes, in relative terms, due to condensation of secondary aerosol which is most pronounced for the small particles. During the photochemical active period of the day (at noon), secondary formation of condensable organics, which might occur faster than that of inorganics, is responsible for the appearance of less hygroscopic Aitken particles than that of 30 nm, which is consistent with the findings presented in previous studies (Mochida et al., 2008). A special case to be considered is the hygroscopic properties of nuclei particles in winter when the nuclei particles appeared to be externally mixed. In fig. 9, the mean diurnal variability of the number size distributions, averaged over time periods when 30 nm are externally (19/01/2017-28/02/2017) and internally mixed, are presented along with the mean number fraction of the moderately hygroscopic mode (i.e. in the case of externally mixed) and the mean GF (i.e. in the case of internally mixed) of the 30 nm particles.  As stated before, the externally mixed nature of the nuclei particles in winter, is the combined effect of the time scale of the aging and mixing processes of fresh emitted and background aerosol at the DEM station. Specifically, the average number size distribution of ambient aerosol revealed strong primary particle emissions (i.e. increased number concentration of nuclei particles) during the traffic rush hours in the morning and in the evening when biomass burning also significantly contribute in aerosol concentrations. It can be seen than the traffic related emissions in the morning are related to lower contribution of moderately hygroscopic particles than the biomass burning related ones. In the summer (absence of biomass burning emissions), traffic remains the major emission source of nuclei particles; these particles are characterized as internally mixed and of lower hygroscopicity than the aged ones.





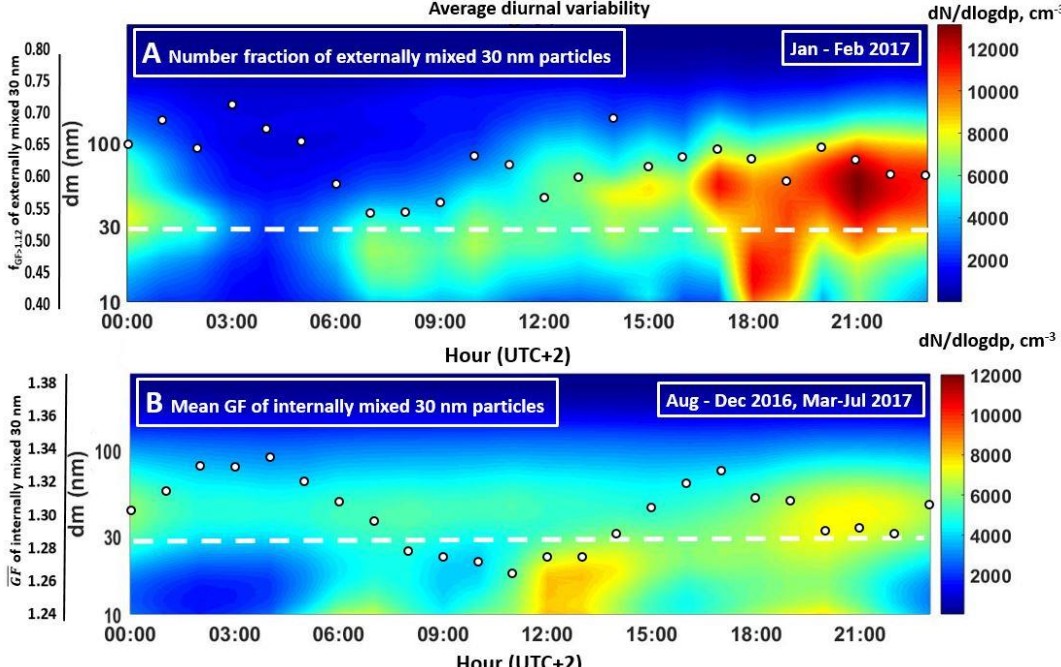

**Figure 9** Diurnal variability of the average number size distributions for periods when (A) 30 nm are externally mixed (19/01/2017-28/02/2017). The white circles represent the number fraction of the externally mixed 30 nm particles with GF > 1.12 and (B) 30nm are internally mixed (August - December 2016, March – July 2017). The white circles represent the mean GF of the internally mixed 30 nm particles.

Figure 10, shows the seasonally resolved diurnal patterns of the GFs of the moderately hygroscopic mode for the specific particle sizes. The 30 nm particles, appeared to have a similar diurnal pattern in all seasons, peaking in late afternoon and early morning. Though, significantly different GFs levels were observed, with the largest GFs found in winter and spring and the lowest in summer. The shape of the diurnal pattern of the larger particles (30, 80 and 250 nm) also depends on season. In the summer and autumn, the GFs were lower within the day, gradually increasing during nighttime (i.e. unimodal daily variability), while in winter/spring a more complex structure was observed.

**Figure 10** Diurnal variation segregated by season of the mean GF of all particles with GF>1.12 for dry diameters A) 30, B) 80 and C) 250 nm.

**3.3 Cluster analysis and aerosol hygroscopic properties**

Cluster analysis was performed on the hourly average size distributions to identify the link between ultrafine particle's hygroscopicity and dynamics. The results are presented in Figure 11A. In addition, the temporal variation of occurrence of each cluster (fig. 11B) was put in context of wind-direction data (fig. 11C), to better interpret the possible sources of aerosol particles. Generally, the number size distributions vary significantly across different regions and environments (Rose et al., 2021). In Athens, secondary aerosol formation, mostly related to sulfate and organics, and traffic-related emissions are the main sources affecting number concentration levels, while biomass burning also consists a major source of aerosol particles





in winter. Specifically, in this study, five clusters were identified which represent the different particle emission sources and formation processes. The modal characteristics of the clustered average number size distribution are summarized in Table S2.

**Figure 11** A) Average number size distribution, B) Hourly frequency of occurrence and C) Frequency of wind direction for each cluster.

**Cluster 1 (Aged traffic mixed with background aerosol)** accounts for 4.2 % of the hourly particle number size distributions. The fact that the frequency of the cluster occurrence peaks at noon and afternoon, while the morning traffic-related peak is missing, together with the prevailing westerly winds, hints to traffic-related pollution transported from the urban area. This 375 was also confirmed from the average number size distribution pattern. The average number size distribution has 3 modes (Table S3); one but less pronounced nuclei mode ($d_p < 10$ nm), a pronounced nuclei mode at 28 nm and a larger Aitken mode at 76 nm. The nuclei mode at 28 nm is probably related to freshly emitted soot particles, internally mixed with other aerosol





species such as organics and sulfates, from the urban sector of Athens. These particles are hygroscopic with mean GF equal to 1.37 (Table S2). The larger Aitken particles represents an externally-mixed aerosol, also composed of a mixture of fresh

(non-hygroscopic) and aged (slightly hygroscopic) particles from different combustion sources (e.g. traffic, biomass burning), the latter of which have undergone aerosol dynamic processing after emission (Brines et al., 2014; Hussein et al., 2014). Generally, the low growth factors are characteristic of polluted air masses (fresh soot from different combustion sources), while the "aging" of the particles makes them more hygroscopic (Vakeva et al., 2002).

**Cluster 2 (urban, nocturnal)** represents 12.1% of the hourly averaged number size distributions. The diurnal frequency of

occurrence profile of this cluster is characterized by an evening peak (21:00 - 00:00 UTC). The average particle number size distribution has 2 modes; the major mode appears in the Aitken size range at 57 nm, while a secondary and less pronounced mode appears in the nuclei size range (< 10 nm). During the evening hours, there is probably a synergetic effect between particle emissions from different combustion sources and the development of the local inversion-nocturnal boundary layer. The 30 nm particles were mostly internally mixed but less hygroscopic than the 30 nm particles of cluster 1, 3 and 5; The

average GF was found equal to 1.28. Only, 8% of these particles were externally mixed with average $GF_{<1.12}=1.07$ and $GF_{>1.12}=1.23$ (wintertime). The 50 nm Aitken particles were externally mixed with average growth factors 1.03, (non or slightly hygroscopic), and 1.21 (moderately hygroscopic).

**Cluster 3 (Nucleation and growth)** accounts for 1.3% of the hourly averaged number size distributions. The diurnal frequency of occurrence profile is characterized by a peak at noon. This cluster was characterized by the least frequency of occurrence

but the highest total number concentration, and occurs almost exclusively under westerly wind directions. The average particle number size distribution appeared to have 3 modes; one nuclei mode at size < 10 nm which contributed more than 60 % to the total number concentration, a second nuclei mode at 16 nm and an Aitken mode at 54 nm. The Aitken mode represents an external mixture of fresh and aged particles from incomplete combustion sources. Whereas, the 30 nm particles belong to larger nuclei mode (16 nm), representing an internally mixed aerosol with mean GF equal to 1.35.

**Cluster 4 (Urban and regional background)** is the most frequent cluster (67%) and represents mainly the contribution of the "regional/urban background aged aerosol", mostly accounting for aged and long-range transported aerosols. The diurnal profile is characterized by an almost stable frequency of occurrence within the day, and minimum total number concentrations (Brines et al., 2014). This cluster is characteristic of the atmospheric processes such as coagulation of ultrafine particles, condensation of gaseous precursors onto pre-existing particles and secondary aerosol formation. The average number size distribution is

composed with 3 modes. The major mode appeared in the Aitken size range (61 nm), a secondary mode appeared in the nuclei size range (< 10 nm) while an additional mode also exists in the accumulation region (175 nm). The 30 nm particles belong to the Aitken mode and appeared to be internally mixed but less hygroscopic than the particles in the cluster 1, 3, and 5; The mean GF was equal to 1.26. Whereas, the 50 and 85 nm particles have similar GFs in all clusters.

**Cluster 5 (Traffic fresh and further growth)** has a frequency of occurrence of 15.3%. This specific cluster is characterized

by a peak in the frequency of occurrence during morning and late afternoon traffic rush hours, while an additional peak appeared at noon. It is not restricted to a specific wind direction, while the size distribution is similar to cluster 1 but with



higher contribution of nuclei particles to the total particle number concentration. The average number size distribution has 3 modes; one nuclei mode at sizes < 10 nm (major mode), probably formed from hot exhaust gases while the cool down and condense (Morawska et al., 2008; Baltensperger 2002), a larger nuclei mode at 14 nm (internally mixed) and one Aitken mode

at 71 nm (externally mixed) having similar hygroscopic properties with the particles in clusters 1 and 3, even though being less hygroscopic especially in the case of the 30 nm particles. This seems reasonable since more oxidized organics and possible slightly more inorganics may contribute to the nuclei particles formed through photochemical driven nucleation compared to those formed through hot engine exhaust gas cooling. The average growth factor of the internally mixed 30 nm nucleation particles (upper tail of nuclei mode) was equal to 1.29, while the average growth factors of the externally mixed 30 nm (4 %)

were $GF_{<1.12} = 1.07$ and $GF_{>1.12} = 1.20$.

**Conclusions**

Hygroscopic properties of ambient aerosol were measured at a suburban environment in Athens, over a period of 12 months, using an HTDMA system for dry particles sizes of 30 nm, 50 nm, 80 nm and 250 nm at relative humidity (RH) of 90%. The

standard deviation σ of the inverted GF-PDF was used as a measure of the mixing state of aerosol. The aerosol was characterized as internally, externally (with two well-defined modes) or continuum of mixing states (with two overlapped modes). In the case of an externally mixed aerosol, the growth factor spectrum was characterized by a non or slightly hygroscopic mode (e.g. black carbon, fresh carbonaceous aerosol) and a moderately hygroscopic mode (e.g. aged traffic, regional or background aerosol).

The data were analyzed with respect to temporal seasonal and diurnal variability; Although the modal GF was rather stable, the contribution of each mode of the GF spectrum to the total hygroscopicity showed a distinct variation. The 30 nm particles were mostly internally mixed and highly hygroscopic, except from wintertime, when a significant fraction of non or slightly hygroscopic mode became distinct. This was attributed to the dominance of primary emissions from different combustion sources generally used for residential heating (e.g. wood burning) and the decline of the aging processes during this period.

The 50 nm and 80 nm Aitken particles were mostly externally mixed except from August when an internally mixed and non or slightly hygroscopic aerosol type dominated all Aitken and larger fractions. The 250 nm particles were externally mixed, with the moderately hygroscopic mode being the major contributor to particle GFs, in all seasons.

The number size distributions were further analyzed by means of cluster analysis to identify the link between aerosol hygroscopicity and aerosol dynamics and origin. The data were categorized into 5 groups representative of two traffic-related

emission sources (fresh and aged traffic), the regional/urban background, the urban-nocturnal aerosol and the photochemically induced nucleation and growth. The mean GF was higher in the case of 30 nm particles, when a nuclei mode between 13 nm and 28 nm appeared in the number size distributions, implying that these particles might have originated from newly formed particles after aging and growth. The hygroscopic properties of the larger Aitken particles and the particles in accumulation



mode do not significantly vary between the different clusters, apart from a slightly higher contribution of the hygroscopic
mode in the case of the "mixed background aerosol" cluster which accounted for the 67 % of the size distributions.
The HTDMA data obtained in this study can be further parameterized and used as a proxy for CCN prediction.

**Acknowledgement**

This research is co-financed by Greece and the European Union (European Social Fund- ESF) through the Operational
Programme «Human Resources Development, Education and Lifelong Learning» in the context of the project "Strengthening
Human Resources Research Potential via Doctorate Research" (MIS-5000432), implemented by the State Scholarships
Foundation (IKY).

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
