# Peer review of "Annual Cycle of Hygroscopic Properties and Mixing State of the Suburban Aerosol in Athens, Greece."

_Atmospheric Chemistry and Physics, 2022_

## Author Comment (AC1)

**Reply to reviewer #1**

We thank the reviewer for the positive feedback and the helpful comments. Below, find our point-to-point response to specific comments.

**Detailed response to comments:**

We present here our response to each comment (blue font) and we quote the respective part of the revised manuscript (grey font).

1. Line 32-33: not only as CCN but also as IN, and those play a role in the indirect effect as well

The ability of aerosol particles to act as both cloud condensation and ice nuclei has been added in the manuscript.

2. Line 48-49: as the kappa theory does not perfectly describe the water activity, you cannot report a certain kappa value for salts either. Kappa is dependent on RH and also particle size, so please say at which RH and D are those kappas are valid or give a range or say that it is an approximation

We agree with reviewer that since the Kappa values are dependent on both RH levels and particle size, we should better give a range instead of certain of kappa values. Therefore, the sentence has been revised as follows:

"The kappa values for highly hygroscopic aerosols, such as salts and sulfates, range between 0.5 and 1.4, for organics from 0.01 to 0.5, whereas for non – hygroscopic aerosol such as soot, the kappa values are close to zero (Petters and Kreidenweis 2007)".

3. Line 59-61: please rephrase the sentence, link between particle hygroscopic growth and what? And you say that there are only a few long-term hygroscopicity studies, there are also quite a few on CCN activity (these also investigate hygroscopicity) please add these as well or say explicitly that you only mean the HTDMA studies here. And are you sure that there are only these long-term HTDMA studies available? Please check again. And by Sellegrini et al. 2014, don't you mean Holmgren et al. 2014? (https://doi.org/10.5194/acp-14-9537-2014)

The sentence has been rephrased as follows:

"Long-term measurements of aerosol hygroscopicity are useful for the better understanding of the link between particle hygroscopic growth and particle emission sources, formation and transformation processes. A few long-term studies of aerosol hygroscopicity and mixing state by means of the HTDMA technique have been published so far, and some of them are mentioned below (Kammermann et al., 2010, Fors et al., 2011, Holmgren et al. 2014)".

4. Line 82-84: is there already a paper on this custom-built HTDMA? If yes, please cite it. If there is not please add more details on the instrument regarding performance and calibration.

The citation "Bezantakos et al., 2013" was added in Line 94.

5. Line 81-92: please add the different flow values for the HTDMA

The aerosol and sheath flow rates of the HTDMA system were added as suggested by the reviewer in Line 81-92, as follows:

"The DMAs were operated with a sheath flow rate of 3.0 L/min, and a monodisperse sample flow rate of ~0.3 L/min".

6. Figure 2: please double check the figure that you name and explain all parts of it: the first item with a nafion has a name: Aerosol, I guess it should be Aerosol dryer, the two DMAs could be labelled as well for better understanding. After the first humidifier and before the RH/T measurement no line is drawn. Half of the textbox of the CPC is missing.

We agree with the reviewer that figure2 was incomplete. Therefore, Figure 2 was redesigned with labels describing all parts of the HTDMA system under field operation.

7. Line 96: "were measured in parallel by the standard SMPS system of the Athens ACTRIS station" was this system moved from the Athens station to the Demokritos station and used there? Or was it operated not exactly where the HTDMA was standing? Not clear from the text.

It is necessary to clarify that the HTDMA was operated in parallel with the SMPS at ACTRIS DEM station, which was also mentioned in the manuscript as "Athens station".

DEM station is the only station in the Athens Metropolitan Area at the suburban location of Ag. Paraskevi and the NCSR Demokritos campus, which submits particle size distribution ACTRIS quality controlled data at the EBAS (nilu.ebas.no), despite the fact that since a few years other ACTRIS stations in Athens have been operating (for aerosol remote and aerosol in-situ data). The following sentence was added, to make it cleared that all the in-situ measurements were performed in parallel at the same site:

"The aerosol number size distributions of ambient aerosol (dry) were measured by means of a Scanning Mobility Particle Sizer system (SMPS) operated at the Athens DEM station."

8. Line 100: RH lower than 45%, is that enough? I always have at least lower than 40% in my mind, but even better if lower than 30%. Please comment on it.

DEM station is a member of ACTRIS/GAW network of stations and it operates following ACTRIS/GAW recommendations in terms of sampling configuration and in-situ instrumentation. The RH levels in the sampling lines is kept below the nominal value of 40% by means of a membrane Nafion dryer. The wrong value of 45% was included in the text by mistake.

9. Line 120: "an inversion algorithm applied to" -> "an inversion algorithm is/was applied to"

The typo corrected as "an inversion algorithm was applied to".

10. Line 122-125: for readers not familiar with the HTDMA inversion, it might be not well understandable. The readers who are familiar with the inversion, it is not necessary. So either explain it better with more details, mentioning at least that when you select a certain size in the first DMA, particles with other sizes go through the DMA as well, that you have multiply charged particles as well and so on… Or leave the whole thing and only refer to the inversion paper, where it is explained in details, and mention that.

The methodology we follow for inverting the HTDMA data is described in detail in the manuscript by Gysel et al. (2009). This citation added as suggested by the reviewer.

11. Line 125: "x2" is it not usually "Chi" and not x?

It can be either both x or Chi. We use "$x^2$ ", as described by Gysel et al., (2009).

12. Line 133-134: The two sentences after each other are repetitions with similar meanings, do you really want both to be there

The sentence was rephrased as follows:

"In the present study, the inverted data can be grouped into three cases, representative of the aerosol mixing state (Fig. 3)."

13. Line 135-137: Typo, bit of too many brackets here?

The typo was corrected.

14. Line 182: some boardening? Or the complete boardening? Based on the average you just cannot tell anything about the mixing state. You could have always perfectly internally mixed aerosol and a changing GF with time which would result in a broad GF-PDF as well. You should clearly state this.

We agree that it should be clearly stated that the broadening can be representative of the degree of mixing of aerosol particles and/or the temporal variability of the GF. Thus, the sentence was rephrased as follows:

"Mean GF-PDFs represent the mean distributions of growth factors and does not necessarily provide a clear picture of the mixing state of these size fractions. More specific, the appearance of a broad mode or two overlapped

modes or two distinct modes does not imply the simultaneously existence of particles of distinctly different hygroscopicity and thus composition but may also result as a matter of temporal GF variation for the long temporal variability data products displayed here. It will be further clarified in the analysis below which factor is prominent at the different cases."

15.Line 187-189: Or another possibility is that BC is simply bigger than 30nm, and therefore whatever is in the nucleation mode is already little bit hygroscopic, and at the bigger sizes one could have the more hygroscopic material condensed on BC cores or even the pure BC particles as well.

We agree that the possibility that BC particles of larger than 30 nm should also be taken into account in most cases, when aging processes quickly move these fresh BC particles to the Aitken mode. This assumption is supported by the fact that during January and February non hygroscopic particles (GF<1.12) are detected inj our suburban aerosol in Athens. We understand your comment is basically in line with this exsplanation and this is now described in more detailed and better clarified is the revised manuscript :

"Non or slightly hygroscopic 30-nm-particles with GF~1.0 are essentially missing, in contrast to particles with D0>30nm indicating that freshly emitted particles, such as bare black carbon, are probably growing faster to sizes larger than 30 nm, and are observed as such largely absent during most of the year with the exception of winter (January & February). Aging processes are not very effective during the dark and colder months, therefore a small fraction of non hygroscopic carbonaceous fresh aerosol is very likely to remain and be detected in these below 30 nm size fraction. It has been found that these aging processes are more efficient for the nuclei mode rather than the higher Aitken modes in modifying their hygroscopicity due to condensation of organics and inorganics onto the pre-existing particles (Vu et al., 2011). Also BC is simply bigger than 30nm, and therefore whatever is in the nucleation mode is already little bit hygroscopic, and at the bigger sizes one could have the more hygroscopic material condensed on BC cores or even the pure BC particles as well."

16. Figure 4: the vertical black line is not defined.

The vertical black line is now defined in fig.4.

17. Figure 4: D=250nm, there is a small peak at high GF of 1.9 or so. Please comment on it, what that could be, if that is a real peak with something highly hygroscopic or just measurement noise?

We agree that as presented in Figure 4, the number fraction of the highly hygroscopic particles in the accumulation mode display a minute peak: We went back to the original and inverted data and we found that the number fraction of particles corresponding to this peak is extremely low and close to zero We therefore consider this as a numerical artefact of the inversion code and not a value with physical significance. We propose to make a note in the manuscript as follows:

"The number fraction corresponding to the minute peak appearing at the high hygroscopicity values for the 250 nm fraction is close to zero and does not appear to be physically meaningful. It can only be considered as an artefact of the inversion code."

18. Line 218: "(fig. 5. Panel A)." Where does this reference belong to? If it belongs to the sentence before: the annual mean is not shown in the figure

The reference "(fig. 5. Panel A)" belongs to the sentence before and therefore the annual mean growth factors were added as annotation in the figure 5.

19. Line 218-219: this sentence is strange: distinct month-to-month variability, but no seasonal variability? What do you mean here? I do see a seasonal variability, For D>30nm higher GFs in spring/late spring, minimum in August then again higher towards the end of the year, and January is again low. 30 looks a bit different with not that pronounced and bit shifted minimum in summer but therefore maybe a higher amplitude of GF change.

We can confirm that this sentence is obviously wrong and was left in the manuscript by mistake due to some copy and paste. Of course there is seasonal variability and we modified the text to describe this finding.

20. Figure 5 and 6: I do not really see the reason to show both figures. To my opinion figure 6 is better suitable to discuss the seasonal changing of the hygroscopicity. And you could add the average values without any problem to the boxplot as well additionally. And make a third column for the kappa boxplot. Here you even see better that there is a seasonal variability in the hygroscopicity to my opinion.

We agree with your suggestion. Therefore, figure 5 was removed from the manuscript. The analysis of seasonal variability of the aerosol hygroscopicity is presented in figure 6, along with the analysis of the kappa values.

21. Line 223-224: monthly average kappa? Are these not the yearly averages?

We confirm that the kappa values presented correspond to the yearly averages. The sentence has been corrected.

22. Line 225: "standard deviation" please change it to GF-PDF standard deviation

Standard deviation was changed to "GF-PDF standard deviation"

23. Line 231-232: sigma for the 250nm particles is as low in September as in August

We agree with the reviewer that sigma for the 250 nm particles is as low in September as in August. The sentence was rephrased.

24. Line 241-242: do you have an idea why February is so much different from January? Why only that month?

February is only markedly different in mixing state. In most size ranges there is a progressive increase in the degree of differences in mixing state from January to February and then a decrease from March onwards. Only in the size range below 30 nm, February strands out as the highest compared to the other months. We have described early that January and February are months when the aging processes are less effective allowing for the different mixing states to appear as such in the suburban aerosol while in all other months and seasons these processes are fast enough to modify aerosol properties like hygroscopicity within the time interval required for the aerosol to spread within the Athens basin from the area of direct emission sources to the background areas.

25. Line 245-246: "Aitken particles and the particles in the accumulation mode ($D_0 > 30$ nm) and the particles in the accumulation mode" too many accumulation modes

The sentence was rephrased as follows:

"In general, the Aitken particles and the particles in the accumulation mode ($D_0 > 30$ nm) can be characterized as an external mixture of moderately hygroscopic (i.e. background aged aerosol) and non-hygroscopic aerosol (i.e. fresh local emissions), respectively."

26. Line 248-257: about the separation of the non- to slightly hygroscopic fraction from the moderately hygroscopic fraction. Selecting a constant GF of 1.12 as a limit means that for the different dry diameters you define a different hygroscopicity limit: for D=30nm GF=1.12 means a kappa of approx. 0.075, the same GF for a 250nm is approx. 0.048. I would suggest to define a kappa limit and calculate a GF limit for each dry diameter. Even if that this would not make a big difference in the results.

The hygroscopic parameter $\kappa$ was calculated as described by Peter and Kreidenweis (2007), by using the mean GF values for each dry size.

$$\kappa = \frac{(GF^3 - 1)(1 - a_w)}{a_w}$$

The mean GF values are the inverted observed values determined by the TDMAinv algorithm by Gysel et al., (2009), before the analysis of the aerosol mixing state and the determination of the non/or slightly and moderately hygroscopic mode.

For the determination of the two hygroscopic modes and the number fraction of each mode, GF = 1.12 was not used as constant value but was the upper limit of the range 0.9 - 1.12 for non-hygroscopic mode. Specifically, two hygroscopic ranges have been selected; a non and/or slightly hygroscopic mode with GF<1.12, and one

moderately hygroscopic mode with GF>1.12. Afterwards, the different integral properties of GF-PDFs (i.e. mean GF and number fraction) were calculated for these GF subranges. The mean GF and the number fraction of non and/or slighlty hygroscopic particles with GF<1.12 is obtained by calculating the mean GF of the subrange of the whole GF-PDF at GF<1.12, according to Eq. (C.9) and Eq. (C.8) in Gysel et al. (2009), respectively. The same procedure was followed and for the moderately hygroscopic mode.

In order to clarify this, the following paragraph was added:

"For the determination of the two hygroscopic modes and the number fraction of each mode, GF = 1.12 is the upper limit of the non-hygroscopic mode. Specifically, two hygroscopic ranges have been selected; a non and/or slightly hygroscopic mode with GF<1.12, and one moderately hygroscopic mode with GF>1.12. Afterwards, the different integral properties of GF-PDFs (i.e. mean GF and number fraction) were calculated for these GF subranges, following the methodology describe by Gysel et al. (2009)."

27. Figure 7, label for f: typo ">/<1.12" should be in subscript as well, not only GF

The label "f" in figure 7 was corrected as: "$f_{GF<1.12}$ / $f_{GF>1.12}$".

28. Line 280-281: "Specifically, for dry particle diameters $D_0 > 30$ nm, the contribution of the non- and/or slightly hygroscopic mode was maximum in spring and minimum in winter." ??? $f_{GF<1.12}$ (Fig 7B) shows something completely different: minimum in spring, higher values in winter, maximum in August.

We agree with the reviewer this description was given in the wrong order by mistake and is now corrected in the text.

"The number fraction of each mode also significantly varied from month to month for all dry sizes, with distinct variability in the relative contributions of particles with small or moderate-to-large growth factors. Specifically, for dry particle diameters $D_0 > 30$ nm, the contribution of the non- or slightly hygroscopic mode was minimum in spring, maximum in August and, in winter"

29. Line 282-283: "In the case of Aitken particles, the non-hygroscopic particles almost equal contributed to aerosol hygroscopicity with the slightly hygroscopic particles in all seasons except for spring." sorry, I do not understand this sentence, or as I can interpret it, that is not seen in the graph, please clarify.

and

30. Line 285-286: "Specifically, the average number fraction of the slightly hygroscopic particles was 0.62, 0.80, 0.67 and 0.70 in winter, spring, summer and autumn, respectively." Do you mean here the fraction of the moderately hygroscopic particles? Please check the naming in the complete discussion on Figure 7, it is very hard to follow this discussion. Maybe it is only coming from the confusion with the names of the different fractions.

We acknowledge again that the description was wrongly given and we update the text as suggested by the reviewer as follows:

"The number fraction of each mode also significantly varied from month to month for all dry sizes, with distinct variability in the relative contributions of particles with small or moderate-to-large growth factors. Specifically, for dry particle diameters $D_0 > 30$ nm, the contribution of the non- and/or slightly hygroscopic mode was maximum in spring, maximum in winter and minimum in August. For particles with $D_0 = 250$ nm, the moderately hygroscopic particles clearly dominate over those with GF<1.12 for all seasons. Specifically, the number fraction of the moderately hygroscopic particles with $D_0 = 250$ nm, was 0.62, 0.80, 0.67 and 0.70 in winter, spring, summer and autumn, respectively."

31. Section 3.2: you only show average values for the diurnal variations. A box plot would include much more information here as well, e.g. for the mean GF, or at least add the standard deviations to the plots.

Please note that all this information together will make the plot difficult to read. Therefore, a new plot with the mean diurnal growth factors and the standard deviations was added in the supplementary information.

32. Section 3.2.: check again the naming of the GF<1.12 and GF>1.12 fractions! It is mixed up again at a lot of places, naming the fraction with GF<1.2 sometimes non-hygroscopic, sometimes non- or slightly hygroscopic and naming the fraction with GF>1.2 slightly or moderately hygroscopic. Hard to follow this section again.

The modes with GF < 1.12 represent the non/slightly hygroscopic aerosol, whereas the GF > 1.12 represent the moderately hygroscopic aerosol (section 3.2). The naming of the GF<1.12 and GF>1.12 fractions was checked and corrected wherever necessary.

33. Line 324: "whereas the minimum appeared at noon (GF < 1.3) (21:00)" ?? At noon or at 21:00?

Line 324 was corrected as follows:

"whereas the minimum GF (<1.3) appeared at the evening (21:00)".

34. Figure 9: how was the time period of the particles being externally or internally mixed exactly defined? And is this plot then showing really only the particles that were externally mixed (A) or internally mixed (B)? Or is it showing just the average for the mentioned time period, when you say, that mostly the particles were externally/internally mixed? Please be more specific here! And why do you show different things with the white circles in panel A and B?

The time period of the externally or internally mixed fractions was not predefined. We examined our database on a monthly basis. We define three different cases of mixing states the internally mixed ($\sigma \leq 0.07$), continuum of mixing states ($0.07 < \sigma < 0.15$) and externally with distinct modes ($\sigma \geq 0.15$) for the monthly GF-PDFs. The classification of the different months as internally or externally mixed (with the continuum of mixing dates classified within the internally mixed case) was based on this analysis.

This way of classifying the seasonal behaviour of the mixing state was also supported by the number fractions of the two mixing states grouped on a monthly basis. It was found that the number fraction of externally mixed particles was lower than 10% in all cases apart from January and February. This latter phrase we will include it in the text in order to reply to this comment and be more informative for the reader.

35. Figure 10: Is there a reason why you only show the seasonally separated diurnal variation of the moderately hygroscopic fraction? And not for the average GF or for both fractions? If yes, please clarify!

And

36. Line 351: "The shape of the diurnal pattern of the larger particles (30, 80 and 250 nm)" only 80 and 250 nm??

And

37. Figure 10: is the difference for both 80 and 250nm particles in the different seasons really significant? And also the diurnal variation? For me these curves look quite flat and similar in each season. Showing not only the average but rather a boxplot or standard deviations or doing some statistical test would help to decide on that.

Comments 35-37 refer to figure 10, and are discussed together, bellow.

As shown in Figure 8 there is very little diurnal variability in the GFs of the non-hygroscopic mode. We also omit the 50 nm as they have the same behaviour as 80 nm. The non or slightly hygroscopic mode is characterized by almost hydrophobic particles, with mean GF close to one. Therefore, we consider suitable to investigate the seasonal diurnal variation of the mean GF of the moderately hygroscopic mode. The results show similar seasonal diurnal variation patterns without significant differences. This is probably indicative of the background aerosol studied in the present work. Since we have a strong indication of the reviewers to reduce the size of the manuscript, and indeed Figure 10 provides similar information as the previous figures, we chose to omit this figure in the revised manuscript. In that way, the length of the manuscript will be reduced without missing important information. The text in the manuscript will be modified accordingly.

38. Section 3.3: the same question which I have asked in the methods part (comment 7), were the SMPS measurements performed at the same station or in Athens? If they originate from the same place (the SMPS measured there where the HTDMA), then please ignore this comment, only state that clear, if not, then you cannot use the hygroscopicity data to describe the different size distribution peaks from another place, and with that this complete section is not valid to my opinion. But only then.

DEM station is a station in Athens Metropolitan area (Figure 1). Please see comment 7.

39. Figure 11: why not to include the GF values in this figure instead of having them only in the supplementary?

And

40. Figure 11: it looks for me that the different GF values are quite stable for all clusters, and even the number fractions of the two hygroscopic modes does not vary too much. Is there really a significant difference between the hygroscopicity of the different size distribution clusters? Like GF_50_2 changes between 1.19 and 1.23 if you look at the different clusters. Please provide some analysis, tests there instead of mentioning some selected GF values in the text. An idea would be also to show the average GF-PDFs for the different clusters and compare them. One should see there better if there are differences or not.

Comments 39 and 40, both refer to figure 11 and will be discussed together below.

The mean GF-PDFs were calculated for each cluster and for the different dry particle sizes (30, 50, 80 and 250nm). The GF-PDFs have been included in Figure 11, in order to provide a more detailed insight into the hygroscopic properties and mixing state characteristics of each cluster. Given that the variation of the number fractions of the non/slightly and moderately hygroscopic modes do not significantly vary between the clusters, we decided to present only the mean GFs of each cluster in Table S2. The mean GFs were higher in the clusters 1 and 3, for all dry sizes. These clusters are related with atmospheric conditions favouring new particle formation or transport of nuclei particles from the city centre to the sampling site. These particles are further mixed with the background aerosol. The less hygroscopic particles are related with cluster 4, which is the most frequent cluster (67%) and represents mainly the contribution of the "regional/urban background aged aerosol", mostly accounting for aged and long-range transported aerosols.

Looking at the GF-PDFs, it can be observed that although the mean growth factors of $D_0 > 30$ nm do not vary significantly between the clusters. Overall, clusters 2 and 5, which represent 12.1% and 15.3% of the hourly averaged number size distributions, respectively, have similar GF-PDFs patterns and average GFs values for all dry particle sizes. Clusters 1 and 3, which account only for 4.2 % and 1.3 % of the hourly particle number size distributions, are characterized by more hygroscopic particles compared to the other clusters. The particles of cluster 4, which represent 67% of the averaged number size distributions, have the lower GFs values compared to the other clusters.

The following paragraph was added:

"The mean GF-PDFs were calculated for each cluster and for the different dry particle sizes (30, 50, 80 and 250nm). Overall, clusters 2 and 5, which represent 12.1% and 15.3% of the hourly averaged number size distributions, respectively, have similar GF-PDFs patterns and average GFs values for all dry particle sizes. Clusters 1 and 3, which account only for 4.2 % and 1.3 % of the hourly particle number size distributions, are characterized by more hygroscopic particles compared to the other clusters. The particles of cluster 4, which represent 67% of the averaged number size distributions, have the lower average GF values compared to the other clusters"

41. Supplementary tables: some description is missing there. Like what the different abbreviations mean? Like GF_50_1 and GF_50_2. Please add an exact definition to each value.

The mean GF-PDFs were calculated for each dry size and cluster. The mean GF-PDFs are depicted in Figure 11 and the mean GFs are presented in Table S2.

**Table S2** Mean GF-PDFs per cluster for different dry particle sizes (30, 50, 80 and 250 nm)

| Cluster | $GF_{Ddry=30nm}$ | $GF_{Ddry=50nm}$ | $GF_{Ddry=80nm}$ | $GF_{Ddry=250nm}$ |
|---------|------------------|------------------|------------------|-------------------|
| 1 | 1.35 | 1.16 | 1.19 | 1.23 |
| 2 | 1.28 | 1.11 | 1.13 | 1.22 |
| 3 | 1.34 | 1.17 | 1.19 | 1.26 |
| 4 | 1.26 | 1.10 | 1.11 | 1.20 |
| 5 | 1.29 | 1.11 | 1.13 | 1.21 |

42. Line 458: something went wrong with the formatting of this reference

The format of the reference was corrected.

43. Overall, quite a few sentences are a little bit hard to follow in the manuscript, I was not always sure what the authors meant. It would be nice if a language edit could be done prior to publication.

Language editing was performed prior manuscript publication.

44. You present kappa values in the manuscript but do not discuss them a lot. I would suggest to add a more detailed discussion on kappa values, maybe compare it to what other studies found as well.

In this study, we investigated the aerosol properties in terms of their hygroscopicity and mixing state by means of an HTDMA system. In that context, detailed analysis of the primary parameter measured with the HTDMA, the GF for selected dry particle sizes, was provided. However, the boxplots for the kappa values have been included in Figure 5 and a more detailed analysis of the kappa values has been provided in the revised manuscript.

---

## Author Comment (AC2)

**Reply to reviewer #2**

We thank the reviewer for the insightful comments and for considering the manuscript worth publishing. Below, find our point-to-point response to specific comments.

**Detailed response to comments** (blue font)**:**

**Summary of recommendations for major revision:**

Please note that we have isolated all comments in this summary and provided a response as follows:

The paper misses GF calculations for 100-150 nm diameter. Please give reason for this. I acknowledge the difficulty in these kinds of measurements, but I require an honest explanation of the missing size range, which is very important for cloud condensation nuclei. Otherwise, the reader might think there was a scientific reason to leave these measurements out.

We acknowledge that this is a valid question one may raise in this context. However, as the reviewer indicates there are practical difficulties when implementing HTDMA measurements. One of this is the limitations arising from selecting several representative size fractions to study but at the same time obtaining data in a relatively high time resolution. Having targeted the hourly time interval and the need to have at least three five minute size distributions (this is the length of an SMPS distribution scan) we could have 4 selected sizes (30nm, 50nm, 80nm, 250nm). This selection may appear ad hoc but it reflects are previous experience with the moments of the number size distribution as observed previously (Vratolis et al., 2019). It is confirmed in the cluster analysis appearing in Figure 11, that we cannot detect an individual mode of particles indicating different physical properties in the range in question (100-150 nm). Please observe in this figure that all particles in this range are included mostly in the separate accumulation mode (100-550 nm). And in any case, 80 nm is in the logarithmic scale is very close to 100 nm. We expect that it is now understandable how the selection of these size was made.

The analysis of GF is much too long and should be substantially shortened in number of figures and in the analysis, which requires a substantial new layout of the written text. The paper should be shortened by at least 1/3 of its current size in the number of words, and at least a half of the figures and tables should be removed in the paper and supplementary information. The readability is very hard at the current state and contains repeatability of similar messages (although shown with new types of results and analysis approach).

Following up on these suggestions we have removed Figure 5 and Figure 10 and we have improved the readability of the document by shortening some discussion. However, the reviewer should consider the suggestions by the first reviewer where additional information is required and further clarification of the figures, while extensive figures are suggested to be included. In order to satisfy both suggestions we have moved a considerable part of information in the supplement and expect that the manuscript is now very much improved.

Some of the analysis of the results contains rather speculative discussions on the reason for high or low GFs. A more detailed analysis with trajectory data, wind speed data, and other meteorological data and a detailed look on individual days with particle number size distribution data is needed to reach firmer conclusions. However, without compromising the obligatory shortening of the paper.

Please consider that the explanations provided on the levels of growth factors are based on the literature from previous studies which have been conducted at DEM station and on the specific characteristics of the microclimate of the city of Athens. Athens is a very large metropolitan area (approx. 5 million people) located in a basin where the diurnal pattern of transport and mixing of particle within the basin is responsible for the most of the particles we observe at DEM station in Athens. Considering that no major city or pollution source is located as far as at

least 100 km we do not except that the fine short-lived aerosol number fractions (with the exception of the accumulation mode) can be characterized by back trajectory analysis. Large events like fires and desert dust transport events are the topic of this study. Valuable information can be derived by local meteorological wind speed and wind direction data. Therefore, a more detailed analysis of meteorological data will be performed for a more detailed explanation about the levels of growth factor values and included in the supplement.

**Detailed revision and comments**

Length of paper:

Several of the figures could be removed (as a very good example Figure 10), which also goes for the figures in the supplementary information. For the analysis, one could for example mention the GF for the first time for all diameters at the same time. Then, one could focus on each individual dry particle diameter and summarize the findings around this particle size, and not mention all the different parameters and circumstances around it (GF, GF PDFs, sigma-values, diurnal variation, seasonal variation, less, intermediate, more hygroscopic modes, relative number fraction of the different hygroscopic modes, meteorological influence, and so on) if it doesn't give new substantial information. Alternatively, the authors can choose another strategy as well for the shortening of the text and removal of figures. Due to the length of all information, it is very difficult for me as reviewer to make a decision on which figures and analyses to shorten. The authors are more familiar with their own data sets and results, and hence I leave it to the authors to make this prioritization.

Following up this suggestion we have removed figure 5 and figure 10. However, reviewer 1 have requested many additional information to be included especially in the supplement. The supplement does not affect the length of the paper and we have not received any indication by the editor or the journal production that this length is problematic. The second suggestion to remove the different parameters calculated from the inversion of HTDMA data are an essential part of the paper, where this type of analysis allows us to resolve for the first time in a suburban area the state of mixing of the fine aerosol size fraction.

Analysis of GFs:

The reason for some of the interpretation of GFs is sometimes speculative. How is it even theoretically possible that the 30 nm diameter particles are more hygroscopic than the 250 nm particles if they come from traffic exhaust? Normally one would expect very high number fraction of hydrophobic particles from relatively fresh fossil fuel combustion at around 30 nm in an urban area (e.g. Guo et al., 2020, https://www.pnas.org/doi/full/10.1073/pnas.1916366117; Titta et al., 2010, doi:10.1016/j.atmosenv.2009.06.021, Kristensson et al., 2013, https://aaqr.org/articles/aaqr-12-07-oa-0194 and many others). You have provided some context to this, explaining that some of the 30 nm hygroscopic particles might come from new particle formation events and that the traffic exhaust particles are aged. But you have to provide more detailed analysis to be able to come to this conclusion: Trajectory analysis (trajectories can be downloaded for free from the Hysplit site) if the air really comes from Athens and under what weather conditions and how long time it took for the air to arrive to the site from Athens and the photochemistry activity with meteorological parameters (for ageing purposes), and closer look at individual size distributions on individual days to see if it resembles a traffic exhaust particle size distribution, or new particle formation or something else. It is not enough to look at the average size distribution of clusters like in Figure 11, since the averaging of several size distributions might mask the shape of the individual size distributions. Besides, a look on an individual day would reveal if it is a new particle formation event day or not.

We would like to demonstrate why our findings and interpretation is not speculative and clarify these points as well as add additional data and discussion in the manuscript so that we respond to the points raised by the reviewer.

Theoretical calculation for hygroscopicity and activation of particles is not straightforward for complex aerosol populations whereas it in our case there is variable state of mixing (different chemical composition) regarding aerosol hygroscopicity even for the same particle size. It is therefore difficult to observe simplified predictions of growth rates according to particle size like for example the one predicted by the Kelvin effect alone. It is evident

here like in other studies that nuclei mode particles below 40 nm are more hygroscopic than the immediate larger size range of Aitken particles (Holmgren et al. 2014), while for a progressive increase in size the hygroscopicity and growth factors increase accordingly.

We do refer to the chemical composition of 30 nm particles (nuclei mode), lines 290-299, but we do not state that this mode originates exclusively from traffic emissions but it may contain traffic emissions as well local and/or distant nucleating and growing nanoparticles. This is why in January and February we observe a complex state of mixing for this mode. Traffic and other combustion sources emissions may also produce particles of different chemical composition with the time scale of aging governing their hygroscopicity. This conclusion is supported by the studies we refer to for Athens and other urban areas (Wang et al., 2018; Enroth et al., 2018; Swietlicki et al., 2008) as well as recent studies (Kim et al., 2020) and the type of analysis the reviewer requires which has to do with the origin of the aerosol impacting our station. Please note that trajectory analysis by HYSPLIT may not be informative here because it has a resolution of 1º*1º (100 km*100km) while the size of the Athens Metropolitan area is approximately 25 km*25 km.

[Figure]

[Figure]

**CPF at the 75th percentile (=2258)**

We have conducted this type of analysis with local meteorological data as shown in the figure above for the 20-38 nm particle population and we conclude taking into account the map of the area that 75% of the values for this size range arrive from the center of the city. Also take into account that the distance mentioned in the manuscript between Athens city center and DEM station is around 7 km. The transport time within the Athens value at the indicative wind speeds observed are yielding estimated transport time between ½ hours to a few hours. These data provide enough evidence to assume that urban emission are the main source of these particles and adequate for aging is ensured. We would like to include this information in the supplement in order to respond in the above comments. Cluster 4 which includes Aitken and nuclei modes is a prominent size distribution pattern for 67% of the individual number size distributions. We do not find any conflict with these findings when check against individual size distributions. In any case the cluster analysis is able to distinguish the cases the reviewer is referring to. Nucleation events are very rare in the frequent heavy load aerosol present in the Athens urban atmosphere (Cluster 3, 1.4% frequency).

A closer look on all of the clusters in Figure 11 for individual days is also necessary to make correct conclusions. To me it seems that the interpretation of the sources of different clusters and their typical sizes is not correct or highly speculative. Based on the size distribution shape, the diurnal variation and the wind direction doesn't lead to the conclusions about the origin of the clusters. For example, the second cluster is not at all nocturnal, and even seems to be more of a traffic exhaust related cluster than cluster 1 due to the association with morning and evening hours, which could be representative of morning and evening traffic. Maybe, the clustering does not even give a valid representation of different representative aerosol types. Maybe you should consider to abandon this analysis and make a manual analysis instead of how air masses influence the size distributions and in turn the GFs as suggested previously? Another example of inconclusive interpretation is the sub-10 nm diameter particles log-normal modes that you present in Figure 11. You have to take a closer look on the possible sources of this mode: Is it particles from traffic exhaust that have nucleated some time after the emissions? It is probably not primary emitted traffic particles in Athens, because the maximum for such a mode should be significantly higher than 20 nm diameter at the time it reaches the site after 1 hour of ageing or similar. Again, a closer look at size distributions in connection with trajectories could reveal the reason for their appearance. Please also explain cluster 5 in a clearer way, it seems to contain contradictory information about the sub-10 nm diameter mode when speaking about cooling of exhaust gas.

We have to agree that cluster analysis of this very large dataset of size distributions can provide us with a qualitative picture of the dynamics and origins of size distributions in Athens. However, our findings are also based in previous papers (Vratolis et al., 2019; Bousiotis et al., 2021; Tsiflikiotou et al., 2019; Kostenidou et al., 2015) now centered and focused on the discussion regarding aerosol hygroscopicity..

We could go on a lengthy discussion if needed on the origin of these clusters and what they represent and as discussed above we have to focus on the dominant features in order to provide meaningful explanations.

For example cluster two is described as urban nocturnal because for sure it is describing traffic origin particles as indicated by the increase of its occurrence during morning hours, but it has the highest frequency during late evening hours beyond the times of evening traffic and coinciding with all traffic/residential heating and other urban emissions prevailing during the whole of the evening hours. Traffic volumes in Athens drop sharply after 9 pm at night. A discussion for the sub 10 nm particles would not be relevant here and in fact we do not use any findings from the cluster analysis for this mode resolved partly because the smaller particle size range analyzed for hygroscopicity is centered at 30 nm.

We fully agree that primary traffic particles are above 20 nm and a mixture of those plus secondary particles from condensable gaseous species is what constitutes the aerosol type dominating the 30 nm particles we analyze.

In any case we agree to reduce the current discussion on cluster analysis to maximum one page avoiding detailed descriptions of their origin and mainly focusing on the modal structure revealed with respect to the selected four sizes for the hygroscopicity analysis.

Grammar:

Some grammatical improvements can be made, for example: "As a direct effect, aerosol particles interact with solar radiation through light absorption and scattering, inducing a positive or negative radiation forcing". "As a direct effect" sounds strange. Another example: "The hygroscopic properties of atmospheric particles are strongly related to particle chemical composition (Gunthe et al., 2009; Gysel et al., 2007), while they undergo continuous changes over particle lifetime". Why do you write "while" in this sentence? These sentences sound a bit strange, and such are found throughout the paper. This needs to be corrected.

All these grammar mistakes will be corrected.

Comprehension:

Chapter 2.3.3 is hard to understand. I know what you mean, since I have been doing similar things. But, not sure that people who haven't done this before will understand your method approach. Please describe it in a few more sentences to make it clearer.

In principle we agree to describe further this section however this will increase the length of the paper even more. We have cited the papers where this methodology is described in more detail.

---

## Author Response (AR2)

**Reply to reviewer**

We thank the reviewer for considering the manuscript worth publishing. Below, we present our point-to-point response to the reviewer's comments in (blue font) and we quote the respective part of the revised manuscript (grey font).

**Summary of recommendations for minor revision:**

You acknowledge in your review answer to me that "Hindsight, it might have been more ideal to probe at e.g. 30 nm, 60 nm, 120 nm, and 250 nm, at least from a CCN activation perspective". You have to be honest in your paper, and mention this fact in the method section. No one will criticize you for this (knowing you can't change your experiments in hindsight). Instead, the reader will appreciate your honesty, and will understand your choice of selected sizes. Otherwise, it will be impossible to understand your reasoning behind it, and others that follow will not understand and don't know how to plan how they should perform similar measurements in similar environments.

The following paragraph was added:

"One of the limitations of the HTDMA technique for size resolved hygroscopicity measurements arises from selecting several representative size fractions to study but at the same time obtaining data in a relatively high time resolution. In our case, four dry sizes were selected to be studied i.e. 30nm, 50nm, 80nm, and 250nm. Our findings reveal that the 50 nm and 80 nm Aitken particles presented similar hygroscopic properties, whereas larger differences were observed between the Aitken particles and the particles in the accumulation size range i.e. 250 nm. At least from a CCN-prediction perspective, our size selection might not be the optimal one, although the GF-PDFs can be interpolated in time and diameter in between the available measurements to describe the hygroscopic behaviour of the aerosol particles in each size bin of the SMPS, without introducing too much error in CCN predictions as confirmed by previous studies (Kammermann et al., 2010). Alternatively, it might be more ideal the hygroscopic properties of ambient aerosol at the dry diameter, $D_0$, 60 nm and 120 nm, instead of at 50 nm and 80nm, to be investigated given that the size range between 100 and 150 nm is considered very important for CCN studies".

Line 106: "Both, aerosol and sheath, flows". Should be "Both aerosol and sheath flows".

We revised the sentence as suggested:

"Both aerosol and sheath flows"

Line 203: "two distinct modes does not imply the simultaneously existence of particles". Should be "two distinct modes do not imply the simultaneous existence of particles".

The sentence was rephrased as:

"two distinct modes do not imply the simultaneous existence of particles".

Line 206: "with GFs to be ranged between 1.17 and 1.41". Should be "with GFs ranging between 1.17 and 1.41".

The sentence was rephrased as:

"with GFs ranging between 1.17 and 1.41".

Line 210: "In winter, the existence of two modes indicates that probably both fresh, (non and/or slightly hygroscopic), and aged, (moderately hygroscopic), emissions from traffic and other combustion sources, (biomass burning, residential heating), contribute to the 30 nm size fraction". What do you mean? Does the traffic contribute to the non and/or slightly hygroscopic, and biomass burning/residential heating to aged-moderately hygroscopic? You have to write this connection between the two sources and the hygroscopicity explicitly in the manuscript.

and

Line 212: "In general, the aging processes are more efficient for the nuclei mode rather than the higher Aitken modes in modifying their hygroscopicity due to condensation of organics and inorganics onto the pre-existing particles (Vu et al., 2021)". Again, what do you mean? Which condensation belongs to which particle mode? Write it out explicitly.

In the present study, the measured GFs of the 30 nm particles were in the range of values reported in previous studies in urban environments influenced mainly by traffic and/or other combustion-related sources (e.g. biomass burning). The hygroscopic properties of the 30 nm particles differ between these studies; some report non-hygroscopic particles, while other studies report moderate hygroscopicity at this size. In general, direct emissions from different combustion sources, nucleation and condensational growth play a key role at this size, resulting in particles of different chemical composition (Wang et al., 2018; Enroth et al., 2018; Swietlicki et al., 2008, Kim et al., 2020). The relative contribution of the different emissions sources cannot be quantitatively determined in the present study, although it is evident that fresh combustion-generated aerosols (traffic and wood burning) tend to be less hygroscopic than the aged one (Vu et al., 2021). In general, aerosols emitted from traffic sources increase their hygroscopic growth factor during atmospheric ageing, but this increment is much lower than that of biomass burning aerosols (Vu et al., 2021). Moreover, traffic-related sources emit almost pure black carbon, whereas black carbon from wood burning is to some extent internally mixed with co-emitted organics and thus more hygroscopic (Motos et al., 2019). In the present study, the nuclei particles are characterized as "moderately hygroscopic" in all seasons, except from winter. In winter a complex state of mixing was observed indicating that both primary and photochemically aged emissions contribute to this size range.

Thus, the following paragraph was added:

"In wintertime, a complex state of mixing was observed indicating that both fresh (non and/or slightly hygroscopic) and aged (more hygroscopic) combustion-generated nanoparticles (i.e. biomass burning, traffic) contribute to the nuclei mode, with the time scale and efficiency of the aging process governing the final hygroscopic properties and state of mixing (Wang et al., 2018; Enroth et al., 2018; Swietlicki et al., 2008, Kim et al., 2020, Vu et al., 2021). The externally mixed nature of the nuclei mode reflects the less efficient aging and coating of the fresh combustion-related nanoparticles during the dark and cold months of the year."

Line 235: "looking at the GF-PDFs of the particles in the accumulation mode a peak appeared in the highly hygroscopic range. However, the number fraction corresponding to at peak is so low that is unimportant to identify the true nature of this negligible small peak". I understand your reason. But, if this hygroscopic mode is dominating contribution to scattering of solar light for example, then, this statement is not true. Better be a bit careful and write that you have chosen not to focus on this small peak, rather than that it is unimportant.

We understand that the characterization of this small peak as "unimportant" might not be explicitly correct. We went back to the measurement data and we can confirm that this peak whenever presented in the number size distribution spectrum it has a close to zero contribution to the total number concentrations (close to zero number fraction). Therefore, the sentence was revised as follows:

"However, the number fraction corresponding to this peak is extremely low (i.e. close to zero). Thus, we decided not to investigate further the nature of this peak."

Figure 5. Larger fonts needed for figure.

The fonts have been done larger.

Table 1. What do you mean with cold and warm period? Which measurements are from cold and which are from warm periods?

In the table 1, a number of characteristic GFs is presented for different emission sources and aerosol chemical compositions. The table and the title of the table were updated as follows:

**TABLE 1** Mean Growth Factors measured at RH=90% for particles with different chemical composition

| Chemical Composition | Growth Factor, (GF) | Source |
|---|---|---|
| BC, Mineral Dust | <1.05 | Vlasenko et al., 2005 |
| Biomass Burning | 1.15-1.65 | Cocker et al., 2001 |
| Aged wood smoke | 1.3-1.5 | Kotchenruther and Hobbs, 1998 |
| Fresh wood smoke | 1.1-1.3 | Kotchenruther and Hobbs, 1998 |
| Inorganic Ions | ~1.7 | Gysel et al., 2002 |
| Organic Compounds | 1.0-1.7 | Koehler et al., 2006 |
| Fresh traffic emission | 0.92-1.20 | Vu et al., 2021 |
| Aged traffic emission | 1.09-1.29 | Vu et al., 2021 |

Line 285: "The number fraction of each mode also significantly varied". Should read: "The number fraction of each mode was also significantly different".

The sentence was rephrased as "The number fraction of each mode was also significantly different".

Figure S3. You never discuss or present the results for this figure in the text (just mention that standard deviations of GFs can be found in Figure S3). So, you should remove this figure.

Figure S3 removed as suggested by the reviewer.

Line 335: "These data provide enough evidence to assume that urban emission are the main source of these nuclei particles, while adequate time for further aging is also ensured. As the particles undergo atmospheric aging their composition changes, in relative terms, due to condensation of secondary aerosol which is most pronounced for the small particles". Which nucleation particles do you mean? The ones between 00 and 05, or the ones at morning, or the ones between 15 and 20, or the ones between 20 and 00? Sorry, I don't get it. You have to rewrite the explanation again.

The above paragraph has been revised as follows:

"For the 30 nm particles, it was observed that the GF of the moderately hygroscopic mode was higher (GF > 1.3), between late evening and early morning (00:00 – 05:00 UTC+2), when the relative humidity appeared to have the maximum values (fig. S2) as well as at early afternoon (15:00 – 20:00 UTC+2). At the DEM station, the 30 nm particles are primarily related to traffic emissions and to a lesser extent to new particle formation (Vratolis, et al., 2019). This was also confirmed in the present study by the cluster analysis of the number size distributions. Moreover, in fig.S3, the CPF (conditional probability function) polar plot of 75th percentile of the total number concentration in the size range from 20 to 38 nm is presented (Carslaw and Ropkins, 2012). It is obvious that these particles are predominately originated from the urban area, under moderate wind speeds. Taking into account that the distance between Athens city center and DEM station is around 7 km, the transport time within the Athens value at the indicative wind speeds observed are yielding estimated transport time between ½ hour to a few hours. These data provide enough evidence to assume that the observed nuclei concentrations reflect a synergetic effect between different combustion-related urban emissions (e.g. fresh traffic-related aerosol from the neighbourhood urban area and further growth) especially during daytime, and the development of the local inversion boundary layer during night-time."

Line 339: "During the photochemical active period of the day, (at noon), secondary formation of condensable organics, which might occur faster than that of inorganics, is probably responsible for the appearance of less hygroscopic Aitken particles than that of 30 nm, which is consistent with the findings presented in previous studies, (Mochida et al., 2008)." Why would the Aitken mode particles be less hygroscopic due to this than the 30 nm particles? Sorry, but I don't get it again. Please rewrite text again to make it become understandable.

We agree with the reviewer that this paragraph is not easily understandable. If the composition remains the same in the nuclei size range as for accumulation and Aitken mode particles, one would expect lower GFs for smaller particles (Kelvin effect). Here, it is evident that nuclei mode particles are more hygroscopic than the somewhat larger particles at the lower end of the Aitken mode, while hygroscopicity increases with particle size from the Aitken to the accumulation size range. This reflects the differences in the chemical composition between the nuclei mode and the lower end of Aitken particles, with the smaller particles to be a mixture of more hygroscopic compounds. Given that inorganics are more hygroscopic than organics, it is expected higher partitioning of the former to the nuclei size range, while the organics may be more crucial for further growth to larger particles.

In short, the following paragraph was added:

"During the photochemical active period of the day, the secondary formation of condensable organics, which might occur faster than that of inorganics, is probably responsible for the appearance of less hygroscopic Aitken particles (Mochida et al., 2008). Specifically, if the composition remains the same in the nuclei size range as for accumulation and Aitken mode particles, one would expect lower GFs for smaller particles (Kelvin effect). Here, it is evident that nuclei mode particles are more hygroscopic than the somewhat larger Aitken particles. This reflects the differences in the chemical composition between the nuclei mode and the lower end of Aitken particles, with the smaller particles to be a mixture of more hygroscopic compounds. Given that inorganics are

more hygroscopic than organics, it is expected higher partitioning of the former to the nuclei size range, while the organics may be more crucial for further growth to larger particles."

Line 378: "frequency of occurrence during morning and late afternoon traffic rush hours, while an additional peak appeared at noon". Should read: "frequency of occurrence during morning and late afternoon traffic rush hours, with an additional peak that appeared at noon".

The sentence was rephrased as:

"frequency of occurrence during morning and late afternoon traffic rush hours, with an additional peak that appeared at noon".

Cluster 2 seems to be more of a wood burning factor, if it is from the urban area. You claim urban (what do you mean with urban? Car traffic?), but you have to motivate whey the Aitken mode particle number size distribution peaks at 60 nm diameter. If this is car exhaust, the particles would need to grow from around 20 nm diameter to 60 nm diameter within only a few hours transport from Athens to your site, which I think is impossible. If you check that cluster 2 appears more often during wintertime, you will have a strong indication that it comes from wood burning. The high night concentrations already indicate that.

If wood burning was the main contributor, one would expect higher frequency of occurrence in wintertime. But this is not our case. The frequency of occurrence of this cluster shows no significant seasonal variability. Therefore, we consider more appropriate to characterize this cluster as "urban background, nocturnal", which reflects the synergetic effect between particle emissions from different combustion sources (aged traffic, wood burning) and the development of the local inversion-nocturnal boundary layer.

Cluster 4 seems to be totally dominated by long-aged particles (from more distant urban areas than Athens), because there is little diurnal variation and the particles are quite large in size. Could you check the wind directions, that it is not coming from Athens, and it would give some proof of that.

We agree with the reviewer. Cluster 4 was revised as follows:

"Cluster 4 (Mixed urban and regional background) is the most frequent cluster (67%), dominated by aged and long-range transported aerosols."

**Cluster 5.** Could one claim that the relatively fresh particles likely come from the neighborhood urban area, since the nucleation mode particles between 10 and 20 nm diameter haven't had time to grow to larger sizes?

We agree with the reviewer. Cluster 5 is representative of "Fresh traffic and further growth". The following sentence was added:

"This cluster represents the relatively fresh particles predominately transported in the receptor site from the neighbourhood urban area."

---

## Author Response (AR3)

**Reply to editor**

*I have one comment regarding the new paragraph starting "During the photochemical active period of the day, the secondary formation of condensable organics, which might occur faster than that of inorganics, is probably responsible for the appearance of less hygroscopic Aitken particles (Mochida et al., 2008)." Here the key is probably not the rate at which the vapors are produced but their relative concentrations. The condensable organics oxidize from relatively higher concentration of a mixture of volatile organic vapors in contrast to sulfuric acid, which is the main inorganic vapor contributing to the growth of nanoparticles. The differences in the condensable vapor concentrations have a larger impact on nucleation mode particles as the Aitken mode particles as they have smaller volume to begin with. Please reformulate this sentence better.*

We thank the editor for his comment. In order to incorporate his suggestion, the following paragraph was added (line 349-357, see manuscript.docs without trackchanges):

"The fact that Aitken particles appeared to be less hygroscopic than nuclei particles (Holmgren et al., 2014), reflect the differences in availability in the atmosphere between the concentrations of inorganic and organic condensable vapours and their relative contribution to the hygroscopic GF of nuclei and Aitken modes. More specifically, inorganics and especially sulphuric acid is the main component for nuclei particles in polluted urban areas (Stolzenburg et al., 2005), but subsequent growth may be affected by the type and concentrations of condensable species. This process has been already described previously in Athens (Petäjä et al., 2007), where higher GFs for 20 nm and 50 nm indicated higher mass flux of soluble material. The decrease in hygroscopicity in the Aitken mode in this study can be potentially attributed to the dominating influence of higher concentrations of non-hygroscopic volatile organic vapours relative to the available condensable soluble mass."